

# Brief communication: Evaluation of the snow cover detection in the Copernicus High Resolution Snow & Ice Monitoring Service

Zacharie Barrou Dumont[1], Simon Gascoin[1], Olivier Hagolle[1], Michaël Ablain[2], Rémi Jugier[2], Germain Salgues[2], Florence Marti[2], Aurore Dupuis[3], Marie Dumont[4], Samuel Morin[4]

[1]CESBIO, Université de Toulouse, CNRS/CNES/IRD/INRAE/UPS, Toulouse, France
[2]Magellium, Ramonville St Agne, France
[3]CNES, Toulouse, France
[4]Univ. Grenoble Alpes, Université de Toulouse, Météo-France, CNRS, CNRM, Centre d'Études de la Neige, Grenoble, France.

*Correspondence to*: Simon Gascoin (simon.gascoin@cesbio.cnes.fr)

**Abstract**: The High Resolution Snow & Ice Monitoring Service was launched in 2020 to provide near real time, pan-European snow and ice information at 20 m resolution from Sentinel-2 observations. Here we present an evaluation of the snow detection using a database of snow depth observations from 1764 stations across Europe over the hydrological year 2016-2017. We find a good agreement between both datasets with an accuracy of 94% (proportion of correct classifications) and kappa of 0.80. More accurate (+6% kappa) retrievals are obtained by excluding low quality pixels at the cost of a reduced coverage (-13% data).

## 1 Introduction

The snow cover area, defined as the spatial extent of the snow cover on the land surface (Fierz et al., 2009), is a key variable in many hydrology, climatology and ecology studies. Earth observation satellites have been used to routinely map the snow cover area at continental scale since the late 1960s (Matson and Wiesnet, 1981). Such observations are increasingly used for meteorological, climate, hydrological, ecosystem and natural hazards applications. The Committee on Earth Observation Satellites has listed nineteen operational remote sensing products which provide information on the spatial extent of the snow cover either as binary (snow/no-snow) or fractional (snow covered fraction of the pixel area) representation. However, most of them have a spatial resolution of 500 m and above, and therefore do not meet a range of user needs both for science and operational applications (Malnes et al., 2015). Previous studies suggest that the spatial scale of variability of snow depth is less than 100 m (e.g. Trujillo et al., 2007; Mendoza et al., 2020). In snow dominated catchments, a fine description of snow cover properties distribution is important to compute snow melt (Freudiger et al., 2017). High resolution snow cover maps reflect the spatial heterogeneity of the snow cover properties and therefore can be assimilated to improve snow water equivalent estimation (Margulis et al., 2016; Baba et al., 2018). High resolution snow cover maps are also critical to understand plant species distribution in alpine and arctic ecosystems (Dedieu et al., 2016; Niittynen and Luoto, 2018). In the disaster management sector, high spatial and temporal resolution snow products down to 50 m resolution were requested by road and avalanches authorities (Malnes et al., 2015). High resolution snow cover maps can also be useful for outdoor activities.

On behalf of the European Commission, the European Environment Agency has commissioned the development and real-time production of the Copernicus High Resolution Snow & Ice products (HRSI), including a snow cover component to address these needs. In particular, this service provides a canopy-adjusted Fractional Snow Cover (FSC) at 20 m resolution along with a cloud and cloud shadow mask and quality flags. The products are derived from Sentinel-2 observations, resulting in a revisit time less or equal to five days. The products are distributed with a maximal latency of 3 hours after the availability of the level 1C product in the Sentinel-2 mission ground segment, which means that they are generally available on the same day as the sensing time. The

products are computed using MAJA (atmospheric correction and cloud detection) and LIS (snow detection and snow fraction calculation) software (Hagolle et al., 2015; Gascoin et al., 2019). The performance of the snow detection with this processing pipeline was previously evaluated over the French Alps and Pyrenees using snow depth records at 120 stations from the Météo-France database (Gascoin et al., 2019). The accuracy (proportion of correct classifications) was 94 % ($\kappa = 0.83$), with a higher false negative rate than the false positive rate. However, this evaluation was spatially limited to 10 Sentinel-2 tiles in France (a tile is 110 km by 110 km), whereas the HRSI products cover 1054 Sentinel-2 tiles over 39 countries in Europe. Any operational snow cover detection algorithm applied to optical multispectral imagery is challenged by spectral similarities between clouds and the snow cover (Stillinger et al., 2019), forest cover obstruction (Xin et al., 2012) and lack of solar irradiance during the winter particularly in mountain regions (due to shading from the surrounding slopes) and high latitude regions (due to low sun elevation). These factors vary significantly across Europe and could have been misrepresented by the former evaluation. In the aim of providing a more robust assessment of the snow product reliability to users of the service, we report here on a much more extensive evaluation using 1764 stations from 36 countries, covering a wider range of climate and topographic conditions. This evaluation was made possible thanks to a massive processing of the Sentinel-2 archive using MAJA and LIS to generate the HRSI collection (about 600'000 products, i.e. 500 Terabytes of input data).

## 2 Data and Methods

### 2.1 In situ data

The evaluation database was prepared by merging two datasets of in situ snow depth (height of snow, HS) measurements. First, we extracted daily snow depth measurements of 1094 SYNOP data (WMO automatic weather station) covering 36 countries. Then, we selected daily data from a recent compilation of snow depth measurements in the Alps (Matiu et al., 2021). The latter dataset consisted of 670 stations located in France, Italy and Germany. The evaluation period spans a hydrological year from 1 Sep 2017 to 31 Aug 2018. This period was chosen to take advantage of the 5-days revisit periodicity reached by the Sentinel-2 mission in Sep 2017 and because the Alps dataset is smaller after 2018. All values were rounded to the nearest centimeter. We combined all these data sources into a single dataset totaling 26933 data points of daily snow depth measurements distributed across 36 countries in Europe (Fig.1). A data point was classified as snow covered if HS was strictly greater than a threshold $HS_0$. We tested the sensitivity of this threshold by calculating the confusion matrix between the FSC products and the reference dataset for 1 cm increments of $HS_0$ from 0 to 10 cm (Klein and Barnett, 2003; Gascoin et al., 2015, 2019).

### 2.2 Snow product

We used the on-ground fractional snow cover (FSCOG) layer but the analysis would be identical with the top-of-canopy layer (FSCTOC) as the canopy adjustment does not change the snow classification (HR-S&I consortium, 2020a). Pixels with value of 205 (cloud or cloud shadow) and 255 (no data) were set to "no data". A pixel was classified as snow if $0<FSC\leq100$ and no-snow if FSC=0. We matched each point of the reference dataset with the nearest pixel of an overlapping FSC product that was acquired on the same day, resulting in a maximal distance of $10\sqrt{2}$ m between the pixel center and the station. If there were more than one matching FSC product on the same day, we selected one whose nearest pixel was neither cloud nor no data. We also assessed the impact of the quality layer on the performance. The QCFLAGS (quality control flags) layer provides bit-encoded quality flags to identify lower quality retrievals e.g. due to low sun elevation, thin cloud cover, surface water (HR-S&I consortium, 2020b). Hence we performed the same analysis as above by excluding all pixels with at least a non-zero quality flag, i.e. QCFLAGS>0.





**2.3 Stratification data**
We stratified the analysis using four external variables: tree cover density, land cover type, elevation and country of measurement.
The tree cover density (TCD) was obtained from Copernicus Land Monitoring Service. It was derived using Sentinel-2 data too
and is available at 20 m resolution with pixel values ranging from 0 to 100%. We used the 2015 product and partitioned the data
into 10 segments of equal TCD range. The land cover was obtained from the Copernicus Global Land Service version 3 (Buchhorn
et al., 2020). We used the 2018 discrete classification map where a pixel's label is the majority label from the fractional cover map.
The classes were regrouped into the following labels: closed or open forest, herbaceous vegetation or wetland, urban, water bodies,
snow and ice, shrubs, moss and lichen, bare and sparse vegetation, cropland, and open sea. The elevation was extracted from the
Copernicus global 30 m digital elevation model. We used it to partition our data into 11 segments. We excluded from the analysis
all pixels that were non-valid in at least one the external dataset, so that the population sizes are equal for each stratification
variable.
**2.4 Metrics**
The comparison between in situ/satellite matchups was performed by computing a confusion matrix and the derived false positive
(FP), false negative (FN), true positive (TP), true negative (TN), recall or fraction of successfully identified positives
(TP/(TP+FN)), precision (TP/(TP+FP)), and kappa coefficient ($\kappa$).
**3 Results**
Figure 2 shows the evaluation of the snow/no-snow detection with in-situ data, and in particular the variation of the kappa
coefficient with the $HS_0$ threshold and corresponding confusion matrices. It indicates a good overall agreement between both
datasets with an accuracy of 94% and $\kappa = 0.80$ at $HS_0 = 0$. The kappa coefficient increases to 0.84 if low quality retrievals are
excluded. The optimal $HS_0$ is equal to 1 cm in both cases and used for the analysis with the stratification data. The false negative
rate is higher than the false positive rate (precision is 93% but recall is 78%). The exclusion of low quality data reduces the total
amount of available data points by 13% and increases the recall (82%) more than the precision (94%), meaning that more false
negative errors are avoided. Figure 3 shows that the best performances ($\kappa > 0.8$) are at locations of "urban", "cropland", "open
forest", "herbaceous vegetation" or "bare/sparse" land cover types. A lower performance ($\kappa \approx 0.6$) is evident for the "closed forest"
and "water body" class. The "shrubs" class has a very low performance ($\kappa \approx 0.1$) but there are only 13 snow values in the in situ
data. The analysis by TCD bins shows that performances tend to decrease as the forest cover increases, in agreement with the lower
accuracy for the "closed forest" land cover type. The snow detection is robust across elevations between 400 m and 2800 m with
kappa values above 0.7, but a higher proportion of false negative between 100 m and 400 m is observed; it is likely related to the
presence of dense forest at low elevation in nordic regions. The performances are also shown for the countries with at least 100
data points. Countries with more than 1000 data points (France, Germany, Italy and Turkey) have kappa scores above 0.75 except
Turkey. Finland and Norway, two high latitude countries and with more than 200 data points each, also have kappa scores equal
or above 0.75. Stratifying the data by month (not shown) indicates that the number of false negatives is highest in December while
the accuracy increases every month from January to April.



## 4 Discussion

The results are in line with the previous evaluation with an accuracy of 94% and a kappa of 0.8 and an optimal snow depth threshold of 1 cm close to the previously reported 2 cm (Gascoin et al., 2019). This value is very low, ten times lower than the one that can be obtained with MODIS data (Klein and Barnett, 2003; Gascoin et al., 2015). This suggests that Sentinel-2 is much more sensitive to thin snow cover due to its higher spatial resolution which reduces the prevalence of mixed pixels. We also find that the proportion of FN is larger than the proportion of FP, indicating that the HRSI snow products are more likely to omit a snow pixel than to falsely classify a pixel as snow covered at the stations locations. This study demonstrates that this effect can be partly attributed to the adverse effect of the forest canopy on snow detection as the number of false negatives is higher in the closed forest land cover type. However, the results also show that this tendency for underdetection is present across nearly all subcategories, suggesting that this limitation is not only due to land cover. The lower performance in winter indicates that it may be a consequence of the low signal-to-noise ratio in Sentinel-2 radiances during the periods of low solar elevation angle.

## 5 Conclusion

This brief communication reports on the performance of the HRSI snow classification based on a year of in situ snow depth data. Although the in situ dataset is unbalanced with about four times more no-snow values than snow values, it is sufficiently large to have thousands of observations in the two categories. It is also well distributed across Europe, as we obtained hundreds of observations in many subcategories (country, land cover, elevation, and tree cover density). This dataset therefore allows drawing more robust conclusions than previously on the performance of the MAJA-LIS algorithm to detect the snow cover. We conclude that Sentinel-2-derived HRSI snow products are sufficiently reliable to study snow cover variations across the variety of European landscapes from the northernmost Arctic regions to the southern semiarid mountains, excluding the densest forest regions. Although the evaluation dataset spans only one year of data, its large geographical scale compensates for its short duration. Further progress would result from a wider public availability of in situ snow cover data in the future over extended periods, including additional sources of data (e.g. citizen science observations, webcam-based snow cover observations, higher resolution satellite observations, etc.).

## Data availability

The FSC products are available from the Copernicus Land website (https://land.copernicus.eu/pan-european/biophysical-parameters/high-resolution-snow-and-ice-monitoring). The TCD product is also available from Copernicus Land (https://land.copernicus.eu/pan-european/high-resolution-layers/forests/tree-cover-density). The SYNOP data are available upon request to the authors. The Alps data providers are Météo France, Deutscher Wetterdienst, Agenzia regionale per la protezione dell'ambiente (ARPA) Friuli Venezia Giulia - Osservatorio Meteorologico Regionale e Gestione Rischi Naturali, ARPA Lombardia, the hydrological office of Bolzano, and Meteotrentino.

## Author contribution

CRediT contributor roles taxonomy. Conceptualization: SG, Data curation: ZBD, MD, Formal analysis: ZBD, Funding acquisition: SG, OH, GS, MA, MD, SM, Investigation: ZBD, SG, Methodology: SG, Project administration: MA, FM, Resources: AD, Software: RJ, GS, OH, ZBD, SG, AD, Supervision: SG, Validation: ZBD, Visualization: ZBD, Writing – original draft preparation: ZBD, SG, Writing – review & editing: ZBD, SG, SM, MD, FM, OH.



**Competing interests**
The authors declare that they have no conflict of interest.
**Acknowledgements**
This work was funded by the European Environment Agency. We acknowledge the Centre National d'Etudes Spatiales in particular
N. Picot and the High Performance Computer team. We also thank M. Matiu for his comments on the manuscript. M.D. has
received funding from the European Research Council (ERC) under the European Union's Horizon 2020 research and innovation
programme (grant agreement No 949516, IVORI).

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




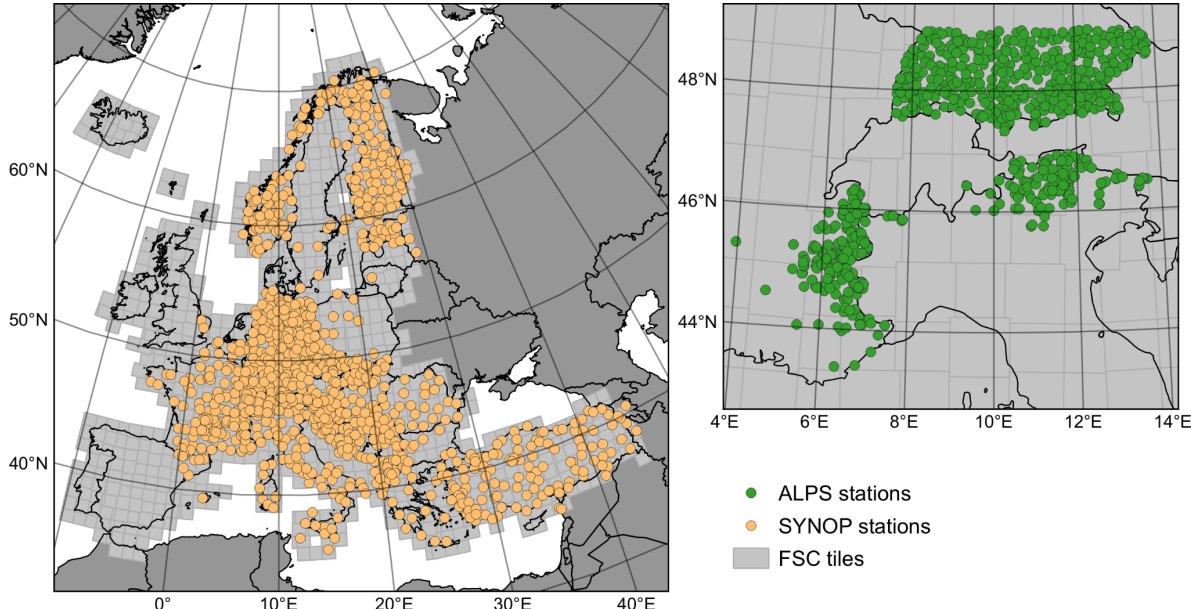


**Figure 1. Map of the study area and location of the in situ measurements. Each FSC (fractional snow cover) tile covers an area of 5490 by 5490 pixels of 20 m resolution.**


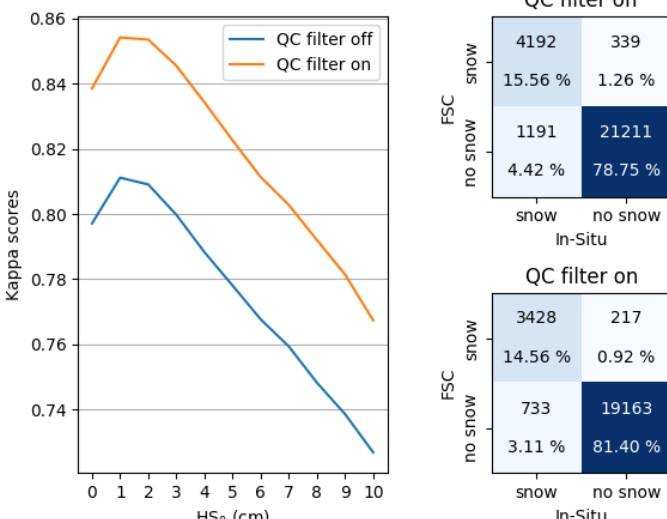


**Figure 2. Evaluation of the snow/no-snow detection with in situ data. Variation of the kappa coefficient with the $HS_0$ threshold and confusion matrices with and without data flagged as low quality. QC filter on/off indicate whether the retrievals were filtered using the corresponding QCFLAGS layer or not.**



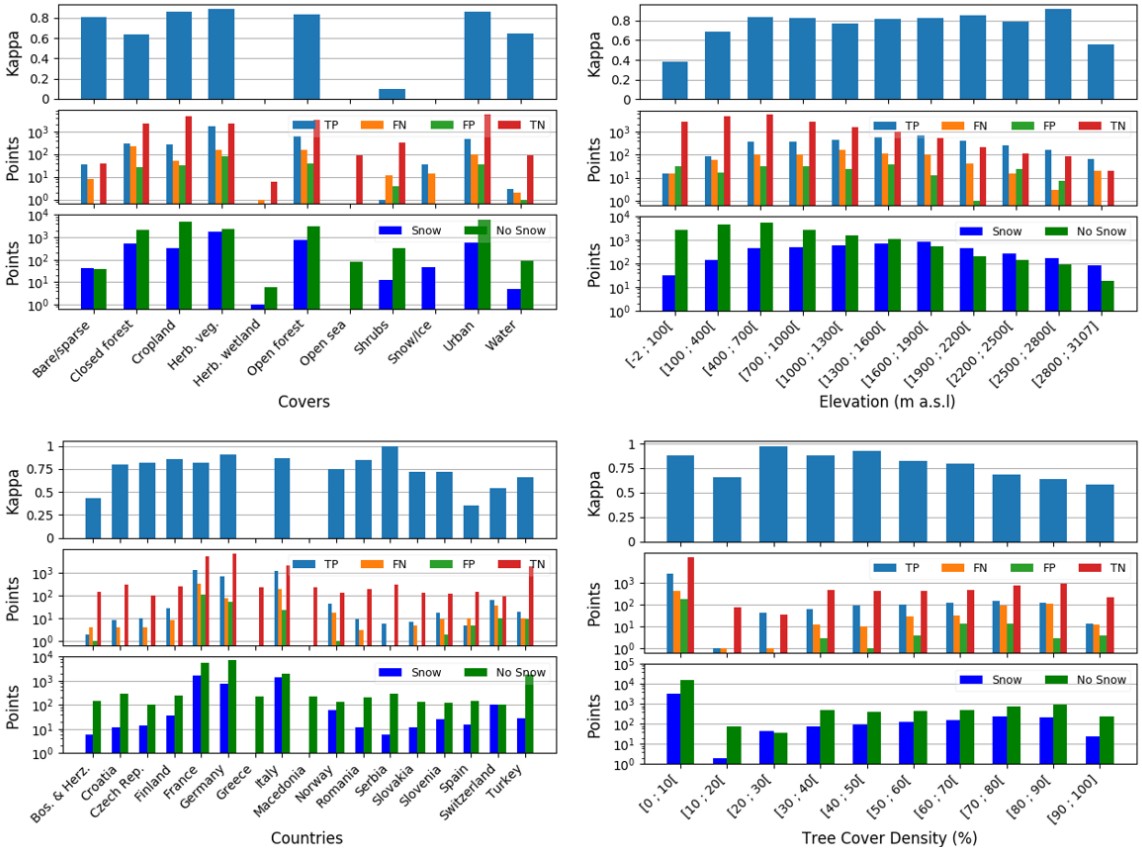

**Figure 3: Results of the evaluation by strata of land cover, elevation, countries and Tree Cover Density. Each subplot shows three histograms for each stratification variable. The histograms represent, from top to bottom respectively, the kappa, the amount of TP (true positive), FN (false negative), FP (false positive) and TN (true negative) on a logarithmic scale and the amount of in situ snow (TP + FN) and no-snow (FP + TN) on a logarithmic scale for each strata. A kappa score of zero happens when there are zero snow observations or zero no-snow observations for either the HRSI FSC or the reference dataset. For example, we get a kappa of zero in Greece despite the results being all true negatives.**