# Peer review of "Copernicus High Resolution Snow & Ice Monitoring Service"

_The Cryosphere, 2021_

## Author Response (AR1)

**Response to referee comment 1**

We would like to thank the referee for the time spent evaluating our work and the relevant suggestions. We provide a point-by-point response below.

1. L84: In the section of the Metrics, the definition of the (overall) accuracy ((TP+TN)/(TP+FP+FN+TN)) should also be added.

We have added this definition.

2. L104: "the number of false negatives is highest in December while the accuracy increases every month from January to April." It is interesting, but 1) "the data" means the data of Finland and Norway ? or the data of all the European countries studied ? 2) What is the cause of the behavior of FN? If possible, please add the figure for the data by month to this report.

We have modified the text to clarify that the data refers to all countries. In the Discussion section (L115) we attribute the behavior of the FN to the lack of direct solar radiation. Given that the number of figure is limited to three for this type of manuscript, we have added the requested figure to a supplementary document (figure shown below). We referred to this figure S1 in the Discussion.

[Figure]

**Figure S1: Results of the evaluation stratified by month. The histograms represent, from top to bottom respectively, the kappa, the amount of TP (true positive), FN (false negative), FP (false positive) and TN (true negative) on a logarithmic**

scale and the amount of in situ snow (TP + FN) and no-snow (FP + TN) on a logarithmic scale. A kappa score of zero happens when there are zero snow observations or zero no-snow observations for either the HRSI FSC or the reference dataset.

3. L108: In the study of Gascoin et al. (2019) the occurrence of false snow detection (i.e., FP) in some large clouds was identified as an issue to be addressed in a future release. However, the FP evaluated in this study seems not to be large compared with FN as shown in Fig. 2. Does this mean that the cloud detection using the MAJA software were improved to eliminate large icy clouds? Or originally the performance of the MAJA is good enough to eliminate icy clouds? The authors may address this issue in the text.

We agree and have added a comment about this in the Discussion.

*The lower proportion of FP than FN in this study also suggests that the occurrence of false snow detection in large clouds that was visually identified in the previous evaluation (Gascoin et al., 2019) is actually not the main issue to focus on in order to improve the product accuracy.*

**Response to referee comment 2**

We wish to thank the reviewer for the positive evaluation of useful comments on our manuscript. We provide a point-by-point response below.

1. The accuracy (proportion of correct classifications) was 94 % (κ = 0.83). Based on the parentheses, one could read that the proportion of correct classifications was 0.83, although it is probably meant that accuracy is just a synonym for the proportion of correct classifications. Please revise the sentence to avoid this risk of misunderstanding.

We have reformulated this sentence to clarify:

*We find a good agreement between both datasets with an accuracy (proportion of correct classifications) of 94% and kappa of 0.80*

2. sensitivity to this threshold?

Indeed, it is probably better to write "to this threshold" instead of "of this threshold". We have made this correction (L62).

3. at least one external data set / at least one of the external data sets?

We have changed to "*at least one of the external datasets*"

4. A reference to the definition of kappa would be useful.

We could cite the paper by Cohen (1960) but we already had the maximum number of references allowed for a brief communication (20). We assumed that the kappa is now well

known in the scientific community and its definition can be easily found in many textbooks or wikipedia.

Cohen, J. (1960). "A coefficient of agreement for nominal scales". Educational and Psychological Measurement. 20 (1): 37–46. doi:10.1177/001316446002000104

1. L88, L106. "Results" should be Section 3 and "Conclusions" Section 4.

In fact we do not understand this comment. The sections are numbered as follows: 3 Results, 4 Discussion, 5 Conclusion. We have separated the Discussion section following the recommendation of the editor after the initial submission.

1. Check the alphabetical order in the list of references.

We have checked and did not notice sorting error (the list was generated automatically with a citation management software)

1. L207-208 (Caption of Fig. 2). Please specify the value of HS_0 used for the right-hand-side matrices.

We have added the information in the caption as suggested.